# Femtosecond Pulsed Fiber Laser by an Optical Device Based on NaOH-LPE Prepared WSe_2_ Saturable Absorber

**DOI:** 10.3390/nano12162747

**Published:** 2022-08-11

**Authors:** Si Chen, Fengpeng Wang, Fangguang Kuang, Shuying Kang, Hanwen Liang, Lijing Zheng, Lixin Guan, Qing Wu

**Affiliations:** 1School of Physics and Electronic Information, Gannan Normal University, Ganzhou 341000, China; 2Heilongjiang Province Key Laboratory of Laser Spectroscopy Technology and Application, Harbin University of Science and Technology, Harbin 150080, China

**Keywords:** WSe_2_, tapered fiber, saturable absorber, an optical device, ultrafast photonics

## Abstract

We report on all-optical devices prepared from WSe_2_ combined with drawn tapered fibers as saturable absorbers to achieve ultrashort pulse output. The saturable absorber with a high damage threshold and high saturable absorption characteristics is prepared for application in erbium-doped fiber lasers by the liquid phase exfoliation method for WSe_2_, and the all-optical device exhibited strong saturable absorption characteristics with a modulation depth of 15% and a saturation intensity of 100.58 W. The net dispersion of the erbium-doped fiber laser cavity is ~−0.1 ps^2^, and a femtosecond pulse output with a bandwidth of 11.4 nm, a pulse width of 390 fs, and a single-pulse capability of 42 pJ is obtained. Results indicate that the proposed WSe_2_ saturable absorbers are efficient, photonic devices to realize stable fiber lasers. The results demonstrate that the WSe_2_ saturable absorber is an effective photonic device for realizing stable fiber lasers, which have a certain significance for the development of potential photonic devices.

## 1. Introduction

The key technology to generate mode-locked ultrashort pulses in ultrafast fiber lasers is the placement of saturable absorption devices in the laser cavity [1,2]. In 2009, Bao et al. proposed that graphene two-dimensional (2D) materials could be used for ultrafast laser mode-locked devices and obtained high-quality graphene mode-locked pulses [3]. Subsequently, research on mode-locked ultrafast fiber lasers with 2D materials has mushroomed. The emergence of new saturable absorbers (SAs) for 2D materials has been shown to create a wave of research in the field of laser technology. The research results have demonstrated that 2D SAs have some unique advantages over conventional commercial semiconductor SAs [3,4], such as broadband saturable absorption, high damage threshold, high modulation depth, etc. [5,6,7]. Currently, 2D SAs, (e.g., topological insulators [8,9], transition metal-sulfur compounds (TMDs) [10,11], MXene [12,13], etc.), have been widely used in fiber lasers to generate ultrashort pulses. Therefore, the multiple functions of 2D materials have driven laser physicists to try to exploit the unique nonlinear optical properties of 2D materials to study the ultrashort pulse generation and soliton dynamics of fiber lasers [14,15,16,17]. Many new 2D nanomaterials are becoming more and more abundant in the optoelectronic research [7,18,19,20,21,22].

To achieve compatibility of 2D nanomaterial SAs with existing fiber laser systems, several structures have been developed for the integration of SAs, such as fiber end face; tapered fiber; D-type fiber; photonic crystal fiber; free-space coupling substrate, etc. The ideal SA material usually has good stability, ultra-fast relaxation time, and a wide wavelength operating range. SA that can maintain long-term stability in the environment is necessary for mode-locked lasers to be able to operate stably. SA of 2D nanomaterials benefits from the third-order nonlinear optical properties of 2D materials flourishing in the field of photonics.

Two-dimensional materials are a kind of material with special morphological characteristics, and the lateral dimension is larger than the thickness dimension (atomic thickness) [23,24]. Therefore, these 2D materials exhibit the 2D planiform nanostructure with a high specific surface area. The discovery of the first 2D material, graphene, opened the door to the two-dimensional world [25]. Although graphene is widely used in various fields, its application in optoelectronics has been limited by its inherent zero bandgap and weak light–matter interactions [26]. Hence, the researchers have been motivated to study the preparation of new 2D materials. After that, the emergence of MoS_2_ compensated for the deficiency of graphene. In contrast, MoS_2_ has continuously tunable band gaps (adjusted by the thickness) and stronger light–matter interactions [27]. Since then, the research on TMDs represented by MoS_2_ has entered a period of prosperous development. As one of the TMDs, WSe_2_ has potential applications in optoelectronic devices, sensing, and nonlinear optics due to its excellent optoelectronic properties, and has been widely concerned since its discovery [28,29,30]. Like MoS_2_, the band gap of WSe_2_ increases with the decrease in thickness, and when the thickness decreases to monolayer, the indirect band will be transformed into a direct band [29]. However, WSe_2_ has many complementary properties to MoS_2_. For example, MoS_2_ is a typical n-type semiconductor, while WSe_2_ is one of the few p-type semiconductors in TMDs. This allows WSe_2_ to be combined with many materials to construct pn-type heterojunctions, broadening its application. In addition, the band gap of monolayer WSe_2_ is 1.6 eV which is smaller than the monolayer MoS_2_ (1.9 eV) [29,31]. Theoretically, the small band gap of the material is more beneficial for the broadband nonlinear optical response. Furthermore, WSe_2_ can generate strongly bound and tunable excitons due to its optically initialized valley polarization and coherence [32]. Moreover, multiple studies have shown that WSe_2_ has an exceptionally strong exciton binding energy [32,33,34]. All the above advantages indicate that WSe_2_ has excellent nonlinear optical properties and has broad application prospects in the fiber lasers [31,35,36,37,38].

WSe_2_ has been extensively studied as an efficient all-optical device for generating Q-switched pulsed lasers. Chen et al. reported a Q-switched fiber laser based on the WSe_2_ PVA films (modulation depth of 3.02%), obtaining a pulse duration of 9.182 μs and a peak power of 720 mW [39]. The CVD-grown WSe_2_ films have been prepared by Liu et al. and these CVD films have been applied to the Q-switched fiber lasers [29]. The modulation depths of 31%, pulse duration time of 4.3 μs, and peak power of 26.7 mW can be obtained by these WSe_2_ Q-switched fiber lasers. Although both Q-switching and mode-locking can obtain short pulses, compared with Q-switching, mode locking can obtain shorter ultrashort pulses, which is mainly related to the SA. However, WSe_2_ has rarely been studied as a SA to generate mode-locked pulsed lasers. Therefore, it is necessary to study the application of WSe_2_ SAs in fiber lasers.

The preparation of WSe_2_ nanosheets and WSe_2_ saturable absorbers are the keys to obtaining high-performance mode-locked pulsed lasers. This is still an important problem for mode-locked fiber lasers. The liquid phase exfoliation (LPE) method has become a general method for the preparation of large-scale, high-quality 2D materials due to its advantages of simple operation and low cost [40,41,42]. However, the production yield by this method is generally low and the obtained WSe_2_ is not stable in other conventional solvents, such as water, thereby hampering the applications of WSe_2_. In this paper, we employed a previously reported NaOH-LPE method, the LPE method based on saturated NaOH solution in N-methyl-2-pyrrolidone (NMP) [43]. The high-yield production of WSe_2_ with a high stability in ethanol can be achieved by this method. According to previous experimental results, the surface of the nanosheets prepared by the liquid phase exfoliation method is usually negatively charged, which enables them to be dispersed in the solution. The OH− ions are absorbed on the surface of WSe_2_, which results in a negative charge, and thus causes excellent stability in water. In addition, ions in NMP solution can intercalate into the interlayers of WSe_2_, thereby enhancing the yield of nanosheets. Furthermore, the combination of the LPE method and the light pulse deposition method has been considered an effective strategy to prepare the 2D materials-based SAs [44].

Here, we demonstrate an erbium-doped fiber pulsed laser based on the WSe_2_ SAs fabricated by NaOH-assisted LPE and optical pulse deposition. The preparation of WSe_2_ is first described and material criteria are performed, followed by the preparation of WSe_2_-SA and verification of the saturable absorption properties of the device. The all-optical device, WSe_2_-SA, has a modulation depth of 15% and a saturable intensity of 100.58 W. The erbium-doped fiber laser achieves an ultrashort pulse output with a spectral width of 11.4 nm and a pulse width of 390 fs. Our work highlights the potential of all-optical devices prepared from WSe_2_ for applications in the field of photonics.

## 2. Material Characterization and Device Fabrication

### 2.1. Material

As shown in Figure 1a, the NaOH-LEP method has been employed to prepare the few-layer WSe_2_ nanosheets. In brief, the bulk phase of WSe_2_ material is placed in a mortar, and the 4 mL NMP solution is added for grinding for 30 min to obtain WSe_2_ powder. Then, the 40 mg WSe_2_ power is added to 20 mL saturated NaOH solution in NMP. After that, the above dispersion liquid of WSe_2_ powder is placed in a sonicator and treated with ultrasonic for 10 h to exfoliate the materials. Notably, the intralayer of the WSe_2_ molecule consists of strong covalent bonds, while the interlayer consists of weak van der Waals forces [45]. This means that the interlayer van der Waals forces of WSe_2_ can be destroyed by the strong ultrasonic process, thereby realizing the purpose of exfoliating WSe_2_ into two-dimensional nanosheets. After sonication proceeding, the WSe_2_ dispersion was centrifuged at 3000 rpm for 20 min to remove the unexfoliated WSe_2_. Then, the sediments with unexfoliated WSe_2_ are removed, and the supernatant contained with few-layer nanosheets is retained. After that, the supernatant is centrifuged at 18,000 rpm for 30 min. The supernatant was removed by pipette and the precipitate was retained. Finally, the retained precipitate is redispersed in ethanol to obtain the dispersion with few-layered WSe_2_ nanosheets.

To confirm the successful preparation of few-layer WSe_2_ nanosheets, some characterization methods have been employed to investigate the morphology of as-prepared WSe_2_ nanosheets, the results are performed in Figure 1. As shown in Figure 1b, transmission electron microscopy (TEM) images show a few very thin and flexible nanosheets with different thicknesses stacked on the other one, and the as-prepared few-layer WSe_2_ nanosheets (lateral dimension is about 600 nm) could be clearly identified with a layered structure. In order to study the thickness and lateral dimensions of the WSe_2_ nanosheets more accurately, the atomic force microscopy (AFM) images of the nanosheets were measured. As shown in Figure 1d, the AFM image shows that size of this WSe_2_ nanosheet is about 1.2 μm. Height profiling of the nanosheet was shown in Figure 1e, which suggests that the thickness of this WSe_2_ nanosheet in Figure 1d marked with line A is 0.8 nm. It is demonstrating that the very thin WSe_2_ nanosheets have been prepared by the NaOH-LEP method. The Raman spectrum is carried out to further investigated the molecular structure of WSe_2_ nanosheets. Additionally, the Raman spectrum is shown in Figure 1c, two characteristic peaks at 248 cm^−1^ and 251.5 cm^−1^ positions can be observed, which corresponds to $E_{2g}^{1}$ (in-plane) and $A_{1g}$ (out-of-plane) modes of WSe_2_. In addition, Raman peaks were observed at 136, 365.7, and 387.6 cm^−1^. It is consistent with the Raman spectra reported in previous work [46,47,48,49]. The above results show that few-layer WSe_2_ nanosheets have been successfully prepared. In addition to this, the absorption spectrum of WSe_2_ is shown in Figure 2, which shows that WSe_2_ has a broadband absorption property, which is beneficial for the application of this material in ultrafast lasers.

### 2.2. Devices

We chose the tapered fiber with simple preparation, large specific surface area and strong evanescent field as the substrate for SA. YOFC SMF 28 fiber is selected, and the tapered fiber is prepared by the hydroxide flame method. Potassium hydroxide (KOH) aqueous solution is used as the electrolyte to electrolyze hydrogen and oxygen in the hydrogen–oxygen machine using the principle of electrolysis of water. Hydrogen is used as an oxygen fuel for fuel and ignition produces a hydrogen–oxygen flame. The outer cladding of the fiber is peeled off, and the outer flame of the hydrogen–oxygen flame is used to cauterize this part of the fiber, and the fiber is pulled at high temperature. The size of the flame is controlled during the preparation process and the fiber is drawn at a uniform speed. A tapered fiber with a diameter of 6 μm, a tapered length of ~600 μm, and a loss of 0.1 dB at 980 nm is obtained.

WSe_2_ for this experiment is prepared by the NaOH-LPE method, which prepared the material with certain homogeneity and facilitated the coupling between WSe_2_ and the near field of the tapered fiber surface. The WSe_2_-SA is prepared by the optical deposition method, and the depositional procedure of the material is observed in real time under the microscope during the preparation project [50,51]. The light source of deposition is a 980 nm pump light source with 50 mW, and a drop of material is first applied to the tapered area of the tapered fiber. With time, a second drop is applied with time, while the loss is observed, and the drop of material is stopped when the loss is 3 dB. The final measured loss of WSe_2_-SA (shown in Figure 3a) is 3.2 dB at 1550 nm and 3 dB at 980 nm. The maximum power (available) that the device can withstand is 2 W of 980 nm pumped light.

The prepared WSe_2_-SA devices are used in the field of lasers with saturable absorption properties in the third-order nonlinear effect, which is an important property to measure as SA. We measured the saturable absorption properties of the devices using a double detection method [51]. The Z-scan can only illustrate the saturable absorption properties of nonlinear optical materials. For the measurement of the saturation absorption properties of SA, the double detection method is simple, economical, and provides a direct indication of the saturation absorption properties of the entire device.

As shown in Figure 3b, the following formula is used to fit the experimental data: Equation $T(I)=1-\Delta T\times \exp(-I/I_{sat})-T_{ns}$, where T(I) is the transmittance, ΔT is the modulation depth, I is the incident peak power, $I_{sat}$ is the saturation power, and $T_{ns}$ is the non-saturated loss. The modulation depth, saturable power, and non-saturated loss of WSe_2_-SA are estimated to be 15%, 100.58 W, and 41.5%. It indicates that WSe_2_-SA has a strong saturable absorption performance at 1550 nm, indicating that the device can be used as an ultrafast optical switch for generating ultrashort pulses at 1550 nm.

## 3. Experimental Setup and Results

The erbium-doped fiber ring laser built in this experiment is shown in Figure 4. YOFC 1013 erbium-doped fiber with group velocity dispersion (GVD) of ~9 ps^2^/km at 1550 nm is selected as the gain medium, and a 980 nm/1550 nm wavelength division multiplexer (WDM) with a pigtail of HI1060 and its length of 1 m, corresponding to a GVD of −7 ps^2^/km. The isolator (ISO) within the cavity is polarization independent to ensure unidirectional propagation. A three-paste polarization controller (PC) regulates the intracavity polarization. The 30% output of the optical coupler (OC) is used to measure the performance of the laser. The performance of the fiber laser is measured with the aid of an 18 GHz high-speed photodetector using a spectrum analyzer (OSA, YOKOGAWA AQ6370C, Janpan), oscilloscope (Agilent MSO7054A, America), RF analyzer (Agilent N9320B, America), and commercial autocorrelator (Femtochrome FR-103, America).

When the 980 nm pump power is set to 285 mW, a stable clamping state can be observed by adjusting the PC. Figure 5a shows a pulse sequence with a time interval of 40.8 ns. The average output power of the laser is 1.03 mW, corresponding to a single pulse energy of 42 pJ. The theoretical calculation corresponds to a cavity length of 8.16 m. The actual cavity length is about 8.3 m, which is consistent with the theoretical value. The GDV of the single-mode fiber in the cavity is −23 ps^2^/km, and the net dispersion in the cavity is calculated to be −0.1 ps^2^. Figure 5b shows the output spectrum with a central wavelength of 1552.68 nm and a 3 dB bandwidth of 11.4 nm. Figure 5c shows the pulse width, and after sech^2^ fitting, the calculated output pulse width is 390 fs, where the calculated time-bandwidth product (TBP) is 0.55. Figure 5d shows a spectral signal-to-noise ratio of ~65 dB at a fundamental frequency of 24.5 MHz, with no harmonics. This indicates that the mode-locked fiber laser has good mode-locking performance. These experimental results confirm that WSe_2_ can be used as a saturable absorbing material to generate stable mode-locked pulse output. In the future, the performance of the mode-locked pulses and the properties of WSe_2_-SA can be customized by continuously optimizing the preparation process of WSe_2_ and improving the design of the laser cavity.

## 4. Discussion

We developed a functional WSe_2_ device, verified the saturable absorption characteristics of the device, and finally used it in an ultrafast fiber laser to realize the output of femtosecond pulses. The earliest graphene-based SA has some excellent nonlinear optical properties, but the zero-bandgap structure and low absorption coefficient of graphene weaken its optical modulation ability, thus limiting the light–matter interaction. Similarly, it is shown that MoS_2_ has layer-dependent nonlinear optical properties, and MoS_2_ has stronger saturable absorption than graphene in the case of a single or few layers.

WSe_2_ is an important member of TMDs materials. Like MoS_2_, it has the advanced properties of strong light–matter interaction and thickness-tunable bandgap. Compared with MoS_2_, WSe_2_ can be more suitably applied to broadband nonlinear optics due to its smaller band gap than MoS_2_. Furthermore, WSe_2_ can generate strongly bound and tunable excitons due to its optically initialized valley polarization and coherence. Moreover, multiple studies have shown that WSe_2_ has an exceptionally strong exciton binding energy. In addition to this, WSe_2_ also has *p*-type electron transport properties that are completely different from MoS_2_.WSe_2_ can be combined with other materials to construct heterojunctions (pn junction) to improve the performance of WSe_2_ optical devices, which is attributed to the inherent *p*-type semiconductor properties and excellent compatibility of WSe_2_. All the above advantages indicate that WSe_2_ has excellent nonlinear optical properties and has broad application prospects in fiber lasers.

So far, the WSe_2_-based implementation of erbium-doped fiber lasers with Q-switched pulses is more common, and mode-locked pulses are less common. According to Table 1, the main preparation methods of WSe_2_ include CVD and LPE, integrated methods include microfiber, tapered fiber, side-polished fiber, and fiber end face (sandwich structure). The few-layer WSe_2_ nanosheets have been prepared by NaOH-LPE method. The larger area and ultrathin WSe_2_ nanosheets with high yield can be achieved by our NaOH-LPE method. Our tapered fiber is more convenient and economical to implement, and most importantly, the damage threshold of the device is up to 1.5W, which is not available in the sandwich structure. It can be seen from the Table 1 that the WSe_2_ thin films prepared by CVD seem to have better properties. This is attributed to the high uniformity of WSe_2_ films prepared by CVD, which is beneficial to obtain better fiber lasers. However, the problems of complex operation, slow growth rate of thin films, unfriendly environments, poor repeatability and high cost limits the application of the CVD method to a certain extent, especially for the preparation of large-scale 2D materials. The simple operation and low cost of LEP make it an effective means for the rapid and large-scale preparation of 2D materials. Moreover, in order to solve the general problem of low yield in the LPE method, saturated NaOH solution was introduced into the system to develop the NaOH-LPE method. In this work, large-scale of WSe_2_ nanosheets with big lateral size and very thin thickness were prepared by the NaOH-LPE method and fabricated as SA. For the SA prepared by the NaOH-LPE method, the parameters obtained in this paper are more optimal [36,52,53].

With the advancement of science and technology, the requirements for stability, repeatability, and easy maintenance of fiber lasers are also increasing. Therefore, one of the important parts of the research from the preparation and structural design of saturable absorbers. Combining WSe_2_ and plasmonic structures to enhance the light–matter interaction is one of the potential strategies to enhance the performance of WSe_2_-SA. Furthermore, designing and constructing SA based on WSe_2_ heterojunctions is one of the other effective strategies. Therefore, we will further carry out the work in the field of WSe_2_-based nonlinear optics around the above two strategies.

## 5. Conclusions

In summary, we achieved ultrashort pulses generated by an erbium-doped fiber mode-locked laser based on WSe_2_-SA. The substrate of the all-optical device WSe_2_-SA is a tapered fiber, and the device has a high damage threshold and a strong saturable absorption property. Based on this all-optical device, we achieved stable mode-locked operation at 1552.68 nm with a pulse of 390 fs and a spectral width of 11.4 nm. The net cavity dispersion of the laser is ~−0.1 ps^2^. This work highlights the potential of WSe_2_ as a member of two-dimensional transition metal materials for applications in photonics.

## Figures and Tables

**Figure 1 nanomaterials-12-02747-f001:**
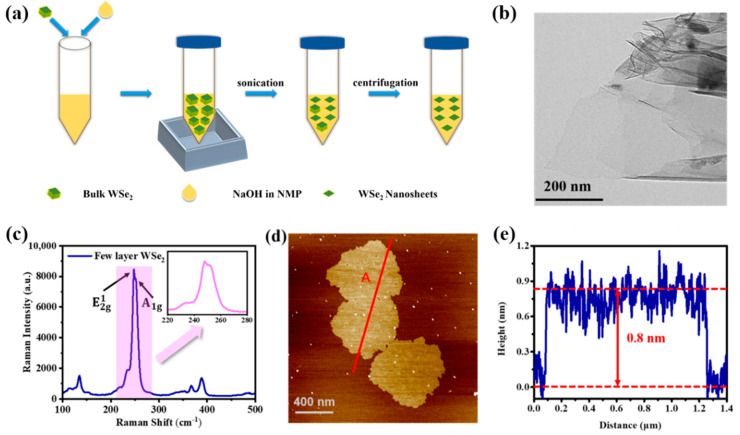
Schematic diagram of fabrication process and characterizations of few-layer WSe_2_ nanosheets: (**a**) Schematic diagram of fabrication process of NaOH-LPE. (**b**) TEM images of the WSe_2_ nanosheets; (**c**) Raman spectrum of few-layer WSe_2_ nanosheet; (**d**) AFM images of few-layer WSe_2_ nanosheet; (**e**) height profiles of the section marked in (**d**).

**Figure 2 nanomaterials-12-02747-f002:**
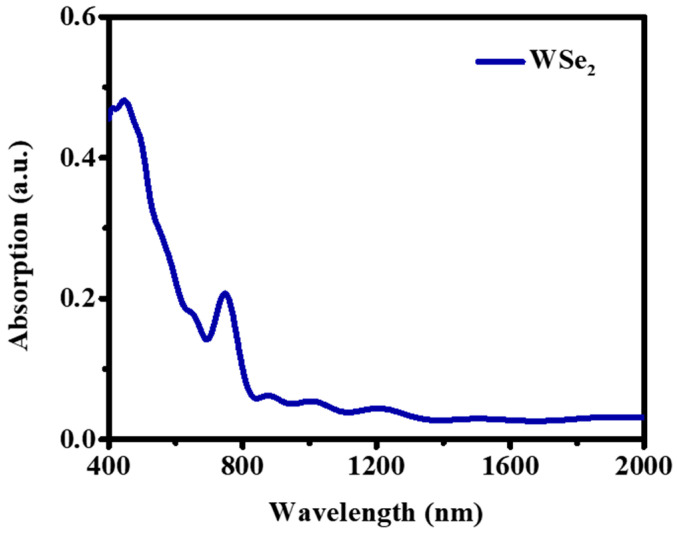
The absorption of few-layer WSe_2_ nanosheets.

**Figure 3 nanomaterials-12-02747-f003:**
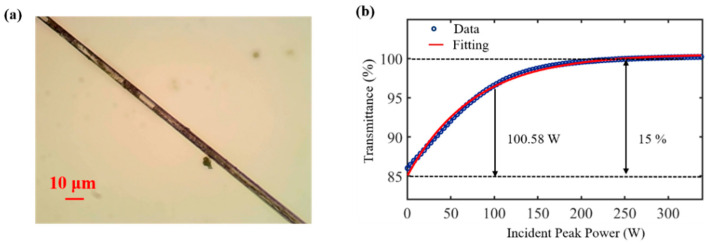
(**a**) WSe_2_-SA: WSe_2_-modified tapered fiber; (**b**) saturation absorption characteristics.

**Figure 4 nanomaterials-12-02747-f004:**
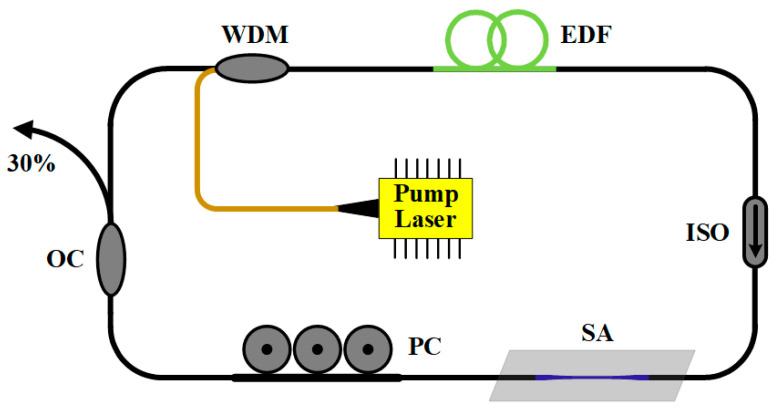
Diagram of erbium-doped fiber mode-locked pulsed laser based on WSe_2_-SA (WDM: wavelength division multiplexer, EDF: erbium-doped fiber, OC: optical coupler, PC: polarization controller, ISO: isolator).

**Figure 5 nanomaterials-12-02747-f005:**
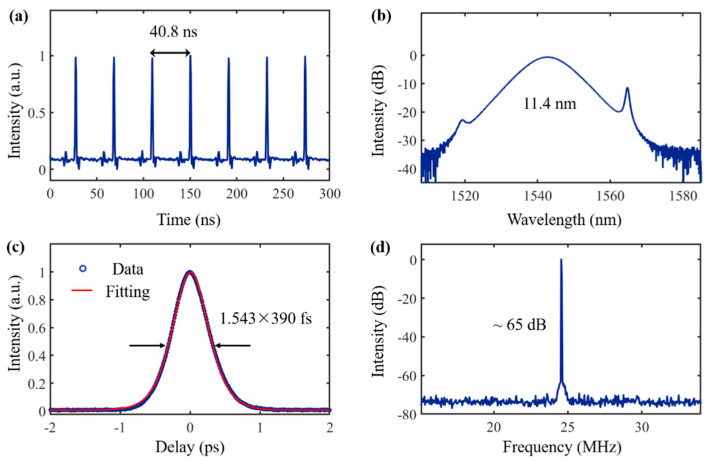
Pulsed output performance of WSe_2_-SA-based erbium-doped mode-locked fiber laser (**a**) typical pulse output sequence with a time interval of 40.8 ns; (**b**) spectrum with a center wavelength of 1552.68 nm and a 3 dB bandwidth of 11.4 nm; (**c**) autocorrelation and sech^2^ fit of the output pulse with a pulse width of 390 fs; (**d**) RF spectrum with ~65 dB signal-to-noise ratio.

**Table 1 nanomaterials-12-02747-t001:** WSe_2_ as a SA for mode-locked fiber laser at 1550 nm.

Fabrication	Integration	*I_sat_ ^a^* (MW/cm^2^)	∆*T ^b^* (%)	Center Wavelength (nm)	Spectral Width (nm)	Pulse Width (fs)	Ref.
CVD	Tapered fiber	15.4	21.89	1557.4	25.8	163.5	[36]
CVD	Microfiber	2.97	8.22	1556.42	6.06	477	[54]
CVD	Fiber and face	14.44	34.41	1558	13.07	185	[52]
LPE	Side-polished fiber	-	0.3	1556.72	7.2	1310	[53]
LPE	Fiber end face	-	0.5	1557.6	2.1	1250	[53]
LPE	Fiber and face	36.28	8.1	1555.2	4.6	698.5	[55]
NaOH-LPE	Tapered fiber	100.58 W	15	1552.68	11.4	390	this work

*^a^* Saturation intensity; *^b^* modulation depth.

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
