# Peer review of "Femtosecond Pulsed Fiber Laser by an Optical Device Based on NaOH-LPE Prepared WSe2 Saturable Absorber"

_nanomaterials, 2022, doi:10.3390/nano12162747_

Round 1

Reviewer 1 Report

The manuscript of S. Chen et al. is reporting on the implementation of a saturable absorber device on the basis of WSe2 nanosheets deposited on a tapered fiber. The device is tested in a fundamental fiber laser setup including a diode pumped Er-doped fiber and the saturable absorber. With the achieved modulation dynamics of 15%, the saturable absorber arrangement enables a clean pulsed operation of the laser emitting pulses with a duration of 390 fs at a total power of approximately 1 mW.

In the present form, the manuscript is well structured and describes all major steps towards the realization as well as the generic properties of the saturable absorber system. The paper cites plenty of references in a correct context, and the presented results are based on experimental evidence. The contents of the paper may be of interest for the scientific community in the fields of ultrashort pulse laser development, nanomaterials and optical technology. In conclusion, the paper represents a comprehensible study with clear results more than worthwhile a publication in Nanomaterials. Prior to publication, the authors may consider some minor aspects for improvement:

·         In line 151 a few more words may be incorporated to describe the applied hydroxide flame method in more detail.

·         Line 167: Even though a reference describing the measurement method for the modulation properties is cited, the authors may give a short summary on the concept and consider possible error margins for the specific application of their SA device.

·         Some remarks on the stability and ageing behavior of the SA-concept may be of interest for the reader.

·         In the present version, the laser system operates at a relatively low power level. The author mention that the power handling capability of their SA-device is relatively strong. Before this background, an estimation of the achievable maximum output power of the system and some remarks on possible thermal management schemes may be interesting for the reader.

·         The present version of the manuscript may be partly improved in respect to the English language style.

Author Response

Comments from the reviewer 1:

1.In line 151 a few more words may be incorporated to describe the applied hydroxide flame method in more detail.

Response: We acknowledge the reviewer for this comment. In the revised manuscript, more descriptions of the hydrogen-oxygen flame method have been added.

“We chose the tapered fiber with simple preparation, large specific surface area and strong evanescent field as the substrate for SA. YOFC SMF 28 fiber is selected and the tapered fiber is prepared by the hydroxide flame method. Potassium hydroxide (KOH) aqueous solution is used as the electrolyte to electrolyze hydrogen and oxygen in the hydrogen-oxygen machine using the principle of electrolysis of water. Hydrogen is used as an oxygen fuel for fuel and ignition produces a hydrogen-oxygen flame. The outer cladding of the fiber is peeled off, and the outer flame of the hydrogen-oxygen flame is used to cauterize this part of the fiber, and the fiber is pulled at high temperature. The size of the flame is con-trolled during the preparation process and the fiber is drawn at a uniform speed. A tapered fiber with a diameter of 6 μm and a loss of 0.1 dB at 980 nm is obtained.”

2.Line 167: Even though a reference describing the measurement method for the modulation properties is cited, the authors may give a short summary on the concept and consider possible error margins for the specific application of their SA device.

Response: We acknowledge the reviewer for this comment. Along with the exploration of nonlinear optical properties of 2D nanomaterials, it is gradually concluded that the double detection method is a simpler method to measure the saturable absorption properties of devices. Although the measurement methods are subject to errors, it is possible to show that the device has saturable absorption properties at 1550 nm using this method. The operation of the double detection method is described in detail in our previous published reference [45, 46]. In the revised manuscript, we have added a summary of this concept.

“The prepared WSe2-SA devices are used in the field of lasers with saturable absorption properties in the third-order nonlinear effect, which is an important property to meas-ure as SA. We measured the saturable absorption properties of the devices using a double detection method [46]. The Z-scan can only illustrate the saturable absorption properties of nonlinear optical materials. For the measurement of the saturation ab-sorption properties of SA, the double detection method is simple, economical, and provides a direct indication of the saturation absorption properties of the entire device.”

3.Some remarks on the stability and ageing behavior of the SA-concept may be of interest for the reader.

Response: We acknowledge the reviewer for this comment. Regarding the stability and aging behavior of the all-optical device based on WSe2, it is mainly the oxidation of the WSe2 material in the air, super humidity and other behaviors that have an impact on the performance of the device. Therefore, we consider the device stability and aging behavior issues from the perspective of device packaging. For example, we will use PDMS to encapsulate the device, or we will dissolve the material in Polydimethylsiloxane (PDMS) in a certain ratio and then encapsulate the tapered fiber to prepare SA. These are the experimental studies that we will conduct later.

4.In the present version, the laser system operates at a relatively low power level. The author mention that the power handling capability of their SA-device is relatively strong. Before this background, an estimation of the achievable maximum output power of the system and some remarks on possible thermal management schemes may be interesting for the reader.

Response: We acknowledge the reviewer for this comment. For the materials-SA of ring fiber lasers, part of the research is based on the "sandwich" structure, for which the damage threshold is very low, generally in the order of mW, which limits its application in high power areas. This limits its application in high power areas. Our proposed WSe2-SA based on tapered fiber structure has a damage threshold of 1.8 W, which is favorable for its application in high power laser cavities, and the maximum output power that can be achieved is what we plan to test later by using nine-cavity hybrid mode locking. For the thermal management of the device, we have studied the thermo-optical modulation achieved based on the photothermal effect of WSe2, and its modulation efficiency is much lower than that of materials such as BP and MXenes.

5.The present version of the manuscript may be partly improved in respect to the English language style.

Response: Many thanks to the reviewer for these valuable comments, and we have checked and revised the English language style of the revised manuscript.

Reviewer 2 Report

The authors report on the realization of a fs pulsed fiber laser by employing as Saturable absorber (SA). WSe2 flakes. There are currently huge efforts in the realization of fs fiber laser source realized by employing as SA nanomaterials.

While the layout of the laser is a standard, the paper can be of interest for researchers working in the field of 2D photonics, but a few revisions are needed.

1.      Something is missed in the structural characterization of WSe2 flaks solution. Figure 1 contains AFM of few flakes, and the line scan shows a profile of 0.8 nm which is compatible with single layer. What is the thickness distribution of the flakes? What is the percentage of single layer flakes?

2.      Raman characterization has low spectral resolution, and a zoom must be provided in the region of 250 cm-1 to shown the two characteristic peaks mentioned in the paper.

3.      The absorption spectrum must also be zoomed in the 400-900 nm region showing the edge (A-eciton peak). Is it possible to infer from the absorption data the average size and thickness of the flakes? See for example Backes, C. et al. Nat. Commun. 5:4576 doi: 10.1038/ncomms5576 (2014).

4.      Details about the device preparation method must be added. The authors just say, “The WSe2-SA is prepared by the optical deposition method….” It is not clear what the optical deposition method stands for. More details are needed or, in alternative, literature references.

5.      Can the author analyze the morphology of the WSe2-SA fiber taper? How many dots per area are deposited onto the fiber taper surface?

6.      What is the origin of the nonlinearities in the WSe2 flakes? Is it thermal effect? Alternatively, electronic?

7.      Does the observed SA depend on the density of flakes deposited on the fiber taper? In addition, can the authors exclude the possibility that the fiber taper by itself has optical nonlinearity? To this aim, a control experiment of saturation absorption of the fiber taper flakes must be performed.

8.      They must show an inset of the panel 5a related to a single pulse (of few ps range), to estimate the after-pulse. From the panel 5a it seems that a dip and few oscillations appear after every single pulse.

9.      What is the threshold for mode-locking operation?

Author Response

Comments from the reviewer 2:

  1. Something is missed in the structural characterization of WSe2 flaks solution. Figure 1 contains AFM of few flakes, and the line scan shows a profile of 0.8 nm which is compatible with single layer. What is the thickness distribution of the flakes? What is the percentage of single layer flakes?

Response: Thanks to the reviewer for the comments. The WSe2 flakes in this manuscript are fabricated by liquid phase exfoliation (LPE) method, which is a general method to prepare various 2D materials such as graphene, MoS2, and black phosphorus. This method is simple and convenient, and can achieve large-scale preparation of 2D materials. However, this method also has certain limitations, that is, the thickness of the nanosheets prepared by the LPE method is difficult to precisely control. Although nanosheets with different thicknesses can be separated by centrifugation at different speeds, nanosheets with a certain thickness distribution are still obtained. It's just that their thickness distribution is more concentrated. Therefore, WSe2 nanosheets of various thicknesses are contained in solution, not all of them are monolayers.

    The thickness distribution of nanosheets is indeed very important for the application of 2D materials. The WSe2 nanosheets achieved by LPE in this manuscript has board thickness distribution. We would also very much like to obtain thickness distribution data and ratios for monolayer nanosheets. However, it is indeed difficult to obtain the thickness distribution of WSe2 nanosheets in solution through a few AFM images. So far, a good solution to this problem is still lacking.

2.Raman characterization has low spectral resolution, and a zoom must be provided in the region of 250 cm-1 to shown the two characteristic peaks mentioned in the paper.

Response: Thanks to the reviewer for the comments. Based on your suggestion, the Raman spectrum of WSe2 has been revised. The revised Figure 1 is shown below, and we have made the corresponding changes in the revised manuscript.

The sentence at line 138-140 has been revised to “And the Raman spectrum is shown in Figure 1b, two characteristic peaks at 248 cm-1 and 251.5 cm-1 position can be observed, which corresponds to  (in-plane) and  (out-of-plane) modes of WSe2.”

  1. The absorption spectrum must also be zoomed in the 400-900 nm region showing the edge (A-eciton peak). Is it possible to infer from the absorption data the average size and thickness of the flakes? See for example Backes, C. et al. Nat. Commun. 5:4576 doi: 10.1038/ncomms5576 (2014).

Response: Thanks to the reviewer for the comments. Based on your suggestion, I have read your attached paper (doi: 10.1038/ncomms5576 (2014)) carefully. This is indeed a nice work and is of great significance for the preparation and application of 2D materials. The absorption of the 2D nanosheet is the sum of the contribution from the edge (E) region and central (C) region of nanosheets. After a series of efforts, a model was established to describe the relationship between the extinction coefficient ratio ExtB/Ext345 and the size (L, x and k) of nanosheets.

equation 4

equation 5

where ExtB and Ext345 are the extinction coefficients at B-excitons and 432 nm, respectively. Modelling the nanosheets as arbitrary 2D shapes with long dimension, L, and aspect ratio, k (k=length/width). Δα=αEC and x is the width of the edge region.

The authors performed extensive TEM and absorption spectroscopy tests on nanosheets of different sizes. This concludes Equation 5 and yields the values of the other parameters in the equation (eg x, αE and αC). Therefore, this established relation can be applied to calculate the average size of MoS2 nanosheets.

For WSe2, its x, αE and αE are all unknowns, which need to be obtained by collecting large number of dates about TEM and absorption spectra. In addition, due to the differences between the preparation methods, the morphologies of the prepared nanosheets are also different due to factors such as ultrasonic time, power and various solutions. This results in a different proportion of the edge area of the nanosheet, which means that the x value will change. Therefore, the above formula cannot be directly applied to the calculation of the size of WSe2 nanosheets.

equation 6

For similar reasons, the thickness of WSe2 nanosheets cannot be calculated directly using Equation 6. It is necessary to measure a large amount of AFM and absorption spectrum data to determine the relationship between the number of layers N and λA.

I also very much hope that the average size and thickness of WSe2 nanosheets can be quickly and easily calculated by this method. However, it does take a lot of time to collect a lot of TEM, AFM and absorption spectroscopy data. In fact, this is not the focus of our work in this manuscript.

In this experiment, saturable absorber device is prepared by loading WSe2 as a functional material onto the surface of tapered fiber. The saturable absorber properties of the whole device are experimentally verified. After a large number of experimental studies, we concluded that the parameters affecting the performance of the whole device include: the diameter of the tapered fiber, the fabrication loss of the tapered fiber, the power of optical deposition, the deposition loss. For different concentrations and volumes of the material, the saturable absorber properties can be achieved in a mode-locked operation as long as the deposition loss is controlled to be around 3 dB.

  1. Details about the device preparation method must be added. The authors just say, “The WSe2-SA is prepared by the optical deposition method….” It is not clear what the optical deposition method stands for. More details are needed or, in alternative, literature references.

Response: Thanks to the reviewer for the comments. References from our previous work ref [45, 46] have also been added to provide more details.

“The WSe2-SA is prepared by the optical deposition method, and the depositional procedure of the material is observed in real time under the microscope during the preparation project [45,46].”

  1. Wu, Q.; Chen, S.; Bao, W.; Wu, H. Femtosecond Pulsed Fiber Laser Based on Graphdiyne-Modified Tapered Fiber. Nanomaterials 2022, 12, doi:10.3390/nano12122050.
  2. Wu, Q.; Jin, X.; Chen, S.; Jiang, X.; Hu, Y.; Jiang, Q.; Wu, L.; Li, J.; Zheng, Z.; Zhang, M.; et al. MXene-based saturable absorber for femtosecond mode-locked fiber lasers. Optics Express 2019, 27, 10159-10170, doi:10.1364/oe.27.010159.

  1. Can the author analyze the morphology of the WSe2-SA fiber taper? How many dots per area are deposited onto the fiber taper surface?

Response: Thanks to the reviewer for the comments.

In this manuscript, tapered fiber and WSe2 are selected for the preparation of WSe2-SA. Firstly, it is because the damage threshold of SA prepared by tapered fiber is large, which is beneficial to the application of SA. Secondly, the SA is prepared by optical deposition method, the power of deposition during the preparation can be controlled, the volume of deposited material can be controlled. The main factors affecting WSe2-SA are the diameter and loss of the tapered fiber, the loss of the optical deposition, and the volume of the deposited material. And we have built up a database of corresponding parameters based on a large number of experiments. The material adsorption on the surface of the tapered fiber can be well observed during the optical deposition process. The thin lamellar morphology of WSe2 nanosheets has been demonstrated in the Figure 1a and Figure 1c. It should be noted that the materials used in this manuscript are WSe2 nanosheets rather than quantum dots. Unlike quantum dots, the average size and thickness of nanosheets in dispersions are difficult to calculate. Therefore, the number of nanosheets that deposited on the fiber taper cannot be accurately counted. Using optical microscopy, the material is loaded onto the tapered fiber surface to form a thin film as shown in Figure 3(a) in the manuscript.

  1. What is the origin of the nonlinearities in the WSe2 flakes? Is it thermal effect? Alternatively, electronic?

Response: Thanks to the reviewer for the comments.

The origin of the nonlinearities in the WSe2 flakes is electrons. A nonlinear medium produces a nonlinear response in the face of high-intensity laser incidence, where the atoms and charges in the medium are displaced relative to each other. , represents the total polarization intensity, represents the electric field, where the dielectric constant is , and  the nth order polarization rate. The electric field strength I is used to represent the refractive index , Many two-dimensional materials have high nonlinear refractive indices, and the study of their nonlinear properties in the field of optics is important for the development of two-dimensional materials in the field of optics.

  1. Does the observed SA depend on the density of flakes deposited on the fiber taper? In addition, can the authors exclude the possibility that the fiber taper by itself has optical nonlinearity? To this aim, a control experiment of saturation absorption of the fiber taper flakes must be performed.

Response: Thanks to the reviewer for the comments.

(1) The optical properties of SA depend on the density of WSe2 flakes deposited onto the tapered fiber, firstly, we choose the WSe2 solution, and secondly, we control the density of the material on the surface of SA through the power control (50 mW of 980 nm pump light) in the deposition process and the volume control (~5-10 μL) of the material nearly. (2) tapered fiber has certain nonlinearity, and we can prove experimentally that the nonlinear effect of tapered fiber cannot achieve the output of mode-locked pulse. (3) When we prepared tapered fiber, we established a database through a large number of experiments and also verified that the output of the pulse is not obtained when only tapered fiber is used for the experiments, while the output of the mode-locked pulse could be obtained for WSe2-SA prepared by optical deposition method, which directly indicates that the contribution of the third-order nonlinear effect of WSe2-SA is greater and the mode-locked pulse is achieved output.

  1. They must show an inset of the panel 5a related to a single pulse (of few ps range), to estimate the after-pulse. From the panel 5a it seems that a dip and few oscillations appear after every single pulse.

Response: Thanks to the reviewer for the comments.

    (1) The pulse output sequence data in figure5a is to show the pulse time interval, because the pulse is very short, what the oscilloscope tested is not the real pulse shape. So figure5c after sech2 fitting the autocorrelation curve, the pulse width is obtained as 390 fs. Therefore, a single pulse from the oscilloscope pulse output sequence is not used in this manuscript to represent the output performance of the laser. (2) The pulse sequence shown in figure5a is well stabilized and the comments made by the reviewer are due to the oscillation of the circuit. The waveform has oscillations with amplitudes that decay quickly after following a rise or fall, a ringing phenomenon that the digital circuit cannot shake off. Most of these phenomena are related to the parameters distributed in the circuit and are difficult to avoid. Still, figure5a can demonstrate the stability of the pulse sequence and its pulse time interval of 40.8 ns corresponding to the fundamental frequency.

  1. What is the threshold for mode-locking operation?

Response: Thanks to the reviewer for the comments.

    The threshold for stable mode-locking operation in this manuscript is 285 mW. The damage resistance threshold of the WSe2-SA device in this manuscript is 1.8 W, and a high damage resistance threshold ensures that the device operates under strong pumping.

Round 2

Reviewer 2 Report

In their revised version the authors bave addressed all the points raised in my report. The paper can be accepted in present form

Reviewer 3 Report

I think that this revised article should be suitable for publication in this journal

This manuscript is a resubmission of an earlier submission. The following is a list of the peer review reports and author responses from that submission.

Round 1

Reviewer 1 Report

The Authors have built and described a mode-locked pulsed laser based on erbium doped fiber and especially prepared WSe2  film saturable absorber which has been deposited on the tapered segment of the fiber. The device seems to work well and basic characteristics have been registered.  I do not have any comments concerning the physics basics, however I have several comments of rather technical character.

First, the paper is written in specific English, there are unneeded repetitions in the text and strange, rarely used words, missing “is” “are” instead of “is”, “was” instead of “were” etc. At any rate the English deserves more careful attention.

In the Abstract and farther the Authors say on “saturation intensity of 100.58 W”. Watts are the units of power, not intensity.

In the “Introduction” chapter there are numerous references to “Professors” (personally) “Academies” etc. This is rather unusual custom in the scientific papers. It would be enough just to refer to original papers of the mentioned persons. In the Introduction the “SA” acronym  used for first time should be defined.

There are several confusing phrases in the chapter 2 (Material characterization and device fabrication);  in the first phrase it is said: “Few-layer WSe2 was exfoliated from their powder...”. Exfoliation is not extraction, something cannot be exfoliated from powder (or liquid). And it should be rather: “...WSe2-s were extracted from their powder..”. In the farther part of this text there is sometimes said on “few-layer WSe2” sometimes on “monolayer”. Decide.

Saying on technology of obtaining the WSe2 saturable absorber in the next part of this description there are next confusing phrases. It should be clearly said than the 980 nm semiconductor laser serving usually as a pump source is in this case used for optical deposition of the WSe2 on the tapered part of the fiber. After firt reading of the text concerning this aspect I was severely confused (it sounded as deposition during pumping or something like that). Here also the word “exfoliation” makes some trouble.

On p. 4 rows 7-5 from down it is: “By fitting Eq. T(I)=..... where.....Tns is the non-saturated loss.”. Then what is going on? This is not logical sentence. The verb is missing.

Farther in this part of text: “...strong saturable absorption performance at 1.5 µm...” Is this the wavelength or some other size?

Summarising, I think that the presented paper can be published in the Nanomaterials, but after taking my comments into account.

Reviewer 2 Report

In my opinion, the manuscript must be modified prior to publication. My main concern is the highly succinct presentation of the results. In fact, the results are described in only one short paragraph (lines 169-180). Moreover, there is no discussion of these results against the background of literature data. The reader is therefore deprived of information about the meaning of the result obtained. Is the obtained result better or worse than other implementations of similar solutions presented in the literature, etc.

Next, I would like to highlight the way the manuscript is presented with little care for the reader. It is challenging to trace the text due to the writing style and the lack of care for English grammar. Therefore, many sentences are difficult or even impossible to understand. For example, the sentence in lines 95-98 is incorrect. Also, the sentences in verses 69-71, 98-99, 124-126, 133-135, and many more. I suggest using the language editing service.

 Reference numbers 31 and 32 are incomplete, and reference number 27 is not correctly formatted.

 Another concern has to do with the style of presenting the background in the introductory part of the manuscript. Especially the paragraph in lines 42-60. The names of universities and scientific institutes appear many times in the text, which in my opinion, has no meaning in the place where the cited results were obtained.

 I believe the manuscript needs major changes and should not be published in its current form.